# Improved log-Gaussian approximation for over-dispersed Poisson regression: Application to spatial analysis of COVID-19

**Daisuke Murakami**[1]*, **Tomoko Matsui**[2]

**1** Department of Statistical Data Science, Institute of Statistical Mathematics, Tachikawa, Tokyo, Japan,
**2** Department of Statistical Modeling, Institute of Statistical Mathematics, Tachikawa, Tokyo, Japan

* dmuraka@ism.ac.jp

## Abstract

In the era of open data, Poisson and other count regression models are increasingly important. Still, conventional Poisson regression has remaining issues in terms of identifiability and computational efficiency. Especially, due to an identification problem, Poisson regression can be unstable for small samples with many zeros. Provided this, we develop a closed-form inference for an over-dispersed Poisson regression including Poisson additive mixed models. The approach is derived via mode-based log-Gaussian approximation. The resulting method is fast, practical, and free from the identification problem. Monte Carlo experiments demonstrate that the estimation error of the proposed method is a considerably smaller estimation error than the closed-form alternatives and as small as the usual Poisson regressions. For counts with many zeros, our approximation has better estimation accuracy than conventional Poisson regression. We obtained similar results in the case of Poisson additive mixed modeling considering spatial or group effects. The developed method was applied for analyzing COVID-19 data in Japan. This result suggests that influences of pedestrian density, age, and other factors on the number of cases change over periods.

## Introduction

Currently, a wide variety of count data are collected through sensors and used for smart urban and regional management (see [1]). For example, in 2020–2021 when the coronavirus disease (COVID-19) spread globally, the daily number of people infected with coronavirus was monitored worldwide, and countermeasures were considered based on the observations [2].

Poisson and other regression models for count data have been used for analyzing the number of COVID-19 cases (e.g., [3, 4]) or other diseases (e.g., [5, 6]). These regression models have also been used in ecology (e.g., [7, 8]), criminology (e.g., [9, 10]), and other fields. Recently, Bayesian Poisson regression, which assumes Poisson distribution for the count data and Gaussian priors for latent variables describing spatial, group, and other effects, is widely used in applied studies.

Still, Poisson regression has remaining issues in terms of (a) computational efficiency and (b) identifiability. Regarding (a), owing to the lack of conjugacy between the Poisson and

**Data Availability Statement:** The R codes used in the "Results: Monte Carlo experiments" and "Results: COVID-19 analysis" sections are available from https://github.com/dmuraka/SimplePoissonApprox_MCsim. The COVID-19

data, which is used in "Results: COVID-19 analysis" section, is owned by JX Press Corporation (https://jxpress.net/ (there is Japanese website only)) and cannot be shared publicly because it is proprietary data. Anyone can purchase the data from the company, and the authors of this study had no special access privileges to the data.

**Funding:** This work was supported by JSPS KAKENHI Grant Numbers 17H02046, JP18H01556, and 18H03628, and JST-Mirai Program Grant Number JP1124793, Japan; all grants were awarded to DM.

**Competing interests:** The authors have declared that no competing interests exist.

Gaussian distributions, an approximate inference is necessary for the estimation. Unfortunately, the Markov Chain Monte Carlo method can be slow for large samples. Faster approximations have been developed for count data regression in a context of additive modeling (e.g., [11, 12]), mixed effects modeling [13], and Gaussian process (e.g., [14, 15]).

Regarding (b), the maximum likelihood estimates of the conventional Poisson regression are unidentifiable or identifiable only weakly for certain data configurations [16, 17], typically, for small samples with many zeros. As we will illustrate later, this property considerably worsens the accuracy of Poisson regression estimates in some cases.

Gaussian approximation is useful for improving (a) the computational efficiency and avoiding (b) the identification problem, which is attributed to the Poisson likelihood [18]. [19–22] proposed closed-form Gaussian approximations for Poisson regression. These approaches are easy to implement, computationally efficient, and free from the identification problem. Given the current situation wherein a wide range of researchers and practitioners use count data, these practical approaches will become increasingly important. Unfortunately, these approximations have the following disadvantages:

i. They have poor approximation accuracy for counts with many zeros as we will demonstrate later. A closed-form approach accurately describing such data is needed.

ii. An arbitrary parameter, which is used to avoid taking the logarithm of zero, must be determined a priori. The value is known to have substantial impact on the modeling result [23]. A closed-form approach without such an arbitrary parameter is needed.

Given that, we develop a log-Gaussian approximation for the over-dispersed Poisson regression that is fast, practical, avoids the identification problem, and overcomes (i)–(ii).

## Methods

### Improved log-Gaussian approximation

**Over-dispersed Poisson regression.**   This study considers the following over-dispersed Poisson model for count variables $Y_i|i \in \{1,\ldots.N\}$:

$$Y_i \sim odPoisson(\lambda_i, \sigma^2), \quad \lambda_i = z_i \exp(\mu_i), \tag{1}$$

where $E[Y_i] = \lambda_i$ and $Var[Y_i] = \sigma^2 \lambda_i$. $\lambda_i$ is a mean parameter, $\sigma^2$ is an over-dispersion parameter, and $z_i$ is a given offset variable.

Suppose that $\mu_i = \mathbf{x}_i'\boldsymbol{\beta}$ where $\mathbf{x}_i$ is a column vector of $K$ explanatory variables and $\boldsymbol{\beta}$ is a vector of regression coefficients. The coefficient estimator and the variance-covariance matrix are given as

$$\hat{\boldsymbol{\beta}} = (\mathbf{X}\prime\boldsymbol{\Lambda}\mathbf{X})^{-1}\mathbf{X}'\boldsymbol{\Lambda}\mathbf{z}, \tag{2}$$

$$Var[\hat{\boldsymbol{\beta}}] = \hat{\sigma}^2(\mathbf{X}\boldsymbol{\Lambda}\mathbf{X})^{-1}. \tag{3}$$

$\mathbf{z} = [z_1,\ldots,z_N]'$ with $z_i = \mu_i + \frac{Y_i - \lambda_i}{\lambda_i}$, $\mathbf{X} = [\mathbf{x}'_1, \ldots, \mathbf{x}'_N]'$, and $\boldsymbol{\Lambda}$ is a diagonal matrix whose $i$-th element equals $\lambda_i$. The coefficients are estimated by an iteratively re-weighted least squares (IRLS) method alternately updating $\hat{\boldsymbol{\beta}}$ and $\hat{\lambda}_i = z_i \exp(\mathbf{x}_i'\hat{\boldsymbol{\beta}})$ until convergence. Given $\hat{\lambda}_i$, the

dispersion parameter is estimated as follows:

$$\hat{\sigma}^2 = \frac{1}{N-K} \sum_{i=1}^{N} \frac{(Y_i - \hat{\lambda}_i)^2}{\hat{\lambda}_i}. \tag{4}$$

The resulting mean estimate $\hat{\lambda}_i$ for the over-dispersed Poisson model is known to be the same as the conventional Poisson regression assuming $\hat{\sigma}^2 = 1$:

$$Y_i \sim Poisson(\lambda_i), \quad \lambda_i = z_i \exp(\mu_i). \tag{5}$$

$\hat{\lambda}_i$ is the Poisson maximum likelihood estimator that suffer from the identification problem as detailed in [17]. Note that the $\lambda_i$ parameter explains not only the mean but also the mode of $Y_i$; for integer-valued $\lambda_i$, $Y_i$ has two modes $\{\lambda_i-1, \lambda_i\}$. Later, we will use the center of the two modes $Mode_c[Y_i] = \lambda_i - 0.5$.

**Log-Gaussian approximation for the Poisson regression.** To overcome the identification problem, we consider approximating the mean estimator $\hat{\lambda}_i$ by using an estimator $\hat{\lambda}_i^+$ obtained from a log-Gaussian model, which is unaffected by the identification problem. The estimated $\hat{\lambda}_i^+$ is used to estimate $\hat{\boldsymbol{\beta}}$, $Var[\hat{\boldsymbol{\beta}}]$, and $\hat{\sigma}^2$.

For the approximation, we need to identify a log-Gaussian model that accurately approximates the Poisson model Eq (5) around $\lambda_i$. Although mean-based log-Gaussian approximations for Poisson regression has been developed (e.g., [20]), the mean and mode of the two distributions behave somewhat differently; the mean and mode of a Poisson distribution are linearly proportional and grow in the same order (and, thus, $\lambda_i$ explains both mean and mode) while those of a log-Gaussian distribution are not linearly proportional, and the mean grows faster than the mode. Therefore, mean-based approximation can have poor approximation accuracy around the mode, which is the distribution center. Considering the success of Laplace or other mode-based approximations in previous studies, it is reasonable to accurately approximate Poisson distribution around the mode.

This study first develops a mode-based closed-form approximation. We will use the Poisson mode center $Mode_c[Y_i]$. Because the mode center is available only when $\lambda_i = E[Y_i] \geq 0.5$ to assure non-negativity, we develop a mode-based approximation for $\lambda_i \geq 0.5$ and another approximation for $\lambda_i < 0.5$. After that, we combine the two approximations for estimating the $\hat{\lambda}_i^+$ parameter.

*Approximation for $\lambda_i \geq 0.5$*

We approximate Eq (5) by using the log-Gaussian variable $y_i$ defined as:

$$y_i + c \sim logN\left(\mu_{i(G)}, \ \frac{1}{\lambda_i + c}\right) \tag{6}$$

where $\mu_{i(G)}$ represents the mean (logscale), and $c$ is a constant required to avoid taking the logarithm of zero. $\frac{1}{\lambda_i + c}$ is an approximate variance for a log-transformed Poisson random deviate.

We perform the approximation so that the mode $Mode[y_i]$ of the log-Gaussian model equals the mode center $Mode_c[Y_i]$ of the Poisson model. The following condition is obtained from the mode matching $Mode_c[Y_i] = Mode[y_i]$:

$$z_i \exp(\mu_i) - 0.5 = \exp\left(\mu_{i(G)} - \frac{1}{\lambda_i + c}\right) - c. \tag{7}$$

Eq (7) suggests that $\mu_i$ and $\mu_{i(G)}$ do not generally have a linear relationship. Exceptionally, they

have the following linear relationship if $c = 0.5$:

$$\mu_{i(G)} = \log(z_i) + \mu_i + \frac{1}{\lambda_i + 0.5}. \tag{8}$$

While existing studies have determined $c$ somewhat arbitrary, $c = 0.5$ is found to be necessary for applying the linear approximation under our assumption.

Let us substitute $c = 0.5$ and Eq (8) into Eq (6). Then, we obtain the following log-Gaussian model approximating the Poisson model:

$$y_i + 0.5 \sim LogN\left(\log(z_i) + \mu_i + \frac{1}{\lambda_i + 0.5}, \ \frac{1}{\lambda_i + 0.5}\right). \tag{9}$$

By organizing Eq (9), we have the following model:

$$\log(y_i^*) \sim N\left(\mu_i, \ \frac{1}{\lambda_i + 0.5}\right), \tag{10}$$

where $y_i^* = \frac{y_i+0.5}{z_i}\exp\left(-\frac{1}{\lambda_i+0.5}\right)$. The log-Gaussian distribution approximates the Poisson distribution around the mode center.

*Approximation for $\lambda_i < 0.5$*

If $\lambda_i < 0.5$, the mode of the Poisson variable $Y_i$ and log-Gaussian variable $y_i^*$ behave somewhat differently: the Poisson mode always takes zero value while the mode of the log-Gaussian distribution gradually converges to zero as $\lambda_i$ (or $\mu_i$) declines. Mode-based approximation is not suitable in this case. Conversely, the means of the two distributions both converge to zero as $\lambda_i$ or $\mu_i$ approaches zero. For $\lambda_i = E[Y_i] < 0.5$, we rely on a mean-based approximation.

By taking the expectation of $y_i^*$ using Eq (10), we have the following relationship:

$$E[y_i^*]\exp\left(-\frac{0.5}{\lambda_i + 0.5}\right) = \exp(\mu_i). \tag{11}$$

Eq (11) implies that, when approximating the Poisson mean function $\mu_i$ (logscale) using $y_i^*$, it should be rescaled by multiplying $\exp\left(-\frac{0.5}{\lambda_i+0.5}\right)$. By applying the rescaling to $y_i^*$, Eq (10) is modified to approximate the mean of the Poisson distribution as follows:

$$\log(y_i^{**}) \sim N\left(\mu_i, \ \frac{1}{\lambda_i + 0.5}\right), \tag{12}$$

where $y_i^{**} = \frac{y_i+0.5}{z_i}\exp\left(-\frac{1.5}{\lambda_i+0.5}\right)$.

**Proposed approximation.** Considering the advantage of the mode-based approximation explained in the "Log-Gaussian approximation for the Poisson regression" section, we use Eq (10) as long as $\lambda_i \geq 0.5$ while Eq (12) is used otherwise. Still, $\lambda_i = E[Y_i]$ is unknown a priori. Considering the property of count data that $P(Y_i < 0.5) = P(Y_i = 0)$, we approximate $P(E[Y_i] < 0.5)$ using the ratio $r$ of zero counts in $\{Y_1, \ldots, Y_N\}$. Given the approximation, Eqs (10) and (12) are applied with probabilities $1-r$ and $r$, respectively. By combining these equations using $r$, our proposed approximation is formulated as:

$$\log(y_i^+) \sim N\left(\mu_i, \ \frac{1}{\lambda_i + 0.5}\right), \tag{13}$$

where $y_i^+ = \frac{y_i+0.5}{z_i}\exp\left(-\frac{1+0.5r}{\lambda_i+0.5}\right)$, which yields Eq (10) if $r = 0$ and Eq (12) if $r = 1$. If all counts

are non-zero, the mode-based approximation is applied for all the samples. As the share of zero counts increases, the mean-based approximation is emphasized.

For the unknown $\lambda_i$ in the variance term, we rely on a plug-in estimator $\hat{\lambda}_i = y_i$. The resulting approximation equation yields

$$\log(y_i^+) \sim N\left(\mu_i, \frac{1}{y_i + 0.5}\right). \tag{14}$$

This plug-in method ignores the uncertainty in the variance term. Consideration of the uncertainty will be an important future task.

Given Eq (14), the estimator for $\mu_i = \mathbf{x}_i'\boldsymbol{\beta}$ yields $\hat{\mu}_i^+ = \mathbf{x}_i'\hat{\boldsymbol{\beta}}^+$ where $\hat{\boldsymbol{\beta}}^+ = (\mathbf{X}'\boldsymbol{\Lambda}_y\mathbf{X})^{-1}\mathbf{X}'\boldsymbol{\Lambda}_y\mathbf{y}^+$, $\mathbf{y}^+ = [y_1^+, \ldots, y_N^+]'$, and $\boldsymbol{\Lambda}_y$ is a diagonal matrix whose $i$-th element equals $y_i + 0.5$. $\hat{\mu}_i^+$ approximates the $\mu_i$ parameter but is free from the identification problem. Thus, we use $\hat{\lambda}_i^+ = z_i \exp(\hat{\mu}_i^+)$ as an estimate of the Poisson mean $\lambda_i$. In other words, the estimated $\hat{\lambda}_i^+$ is substituted into Eqs (2)—(4) to estimate $\hat{\boldsymbol{\beta}}$, $Var[\hat{\boldsymbol{\beta}}]$, and $\hat{v}^2$.

**Proposed approximation for Poisson mixed effects model.** Our approximation is readily extended for (over-dispersed) Poisson mixed effects model (MEM) (e.g., [21]) which is formulated as

$$Y_i \sim odPoisson(\lambda_i, \sigma^2), \quad \lambda_i = z_i\exp(\mu_i), \quad \mu_i = \mathbf{x}_i'\boldsymbol{\beta}, \quad \boldsymbol{\beta} \sim N(0, \boldsymbol{\Sigma_\beta}), \tag{15}$$

where $\boldsymbol{\Sigma_\beta}$ is the variance-covariance matrix for $\boldsymbol{\beta}$. We consider the following estimators for Eq (15):

$$\hat{\boldsymbol{\beta}} = (\mathbf{X}'\boldsymbol{\Lambda}^+\mathbf{X} + \boldsymbol{\Sigma_\beta}^{-1})^{-1}\mathbf{X}'\boldsymbol{\Lambda}^+\mathbf{z}^+, \tag{16}$$

$$Var[\hat{\boldsymbol{\beta}}] = \hat{\sigma}^2(\mathbf{X}'\boldsymbol{\Lambda}^+\mathbf{X} + \boldsymbol{\Sigma_\beta}^{-1})^{-1}, \tag{17}$$

$$\hat{\sigma}^2 = \frac{1}{N-L}\sum_{i=1}^{N}\frac{(Y_i - \hat{\lambda}_i^+)^2}{\hat{\lambda}_i^+}, \tag{18}$$

where $L = tr[(\mathbf{X}'\boldsymbol{\Lambda}^+\mathbf{X} + \boldsymbol{\Sigma_\beta}^{-1})^{-1}\mathbf{X}'\boldsymbol{\Lambda}^+\mathbf{X}]$ is the effective degrees of freedom, $\mathbf{z}^+ = [z_1^+, \ldots, z_N^+]'$ with $z_i^+ = \hat{\mu}_i^+ + \frac{Y_i - \hat{\lambda}_i^+}{\hat{\lambda}_i^+}$, and $\boldsymbol{\Lambda}^+$ is a diagonal matrix whose $i$-th element equals $\hat{\lambda}_i^+$. These estimators are identical to the conventional Poisson MEM if $\mathbf{z}^+$ is replaced with $\mathbf{z}$.

As before, we approximate the Poisson mean $\hat{\lambda}_i^+ = z_i \exp(\hat{\mu}_i^+)$ in Eqs (16)–(18) using the following model approximating Eq (15) around $\lambda_i$:

$$\log(y_i^+) \sim N\left(\mu_i, \frac{1}{y_i + 0.5}\right), \quad \mu_i = \mathbf{x}_i'\boldsymbol{\beta}, \quad \boldsymbol{\beta} \sim N(\mathbf{0}, \boldsymbol{\Sigma_\beta}). \tag{19}$$

Once the Gaussian mixed effects model (Eq 19) is estimated, the approximate Poisson mean $\hat{\mu}_i^+ = \mathbf{x}_i'\hat{\boldsymbol{\beta}}^+$ (logscale) is obtained where $\hat{\boldsymbol{\beta}}^+ = (\mathbf{X}'\boldsymbol{\Lambda}_y\mathbf{X} + \boldsymbol{\Sigma_\beta}^{-1})^{-1}\mathbf{X}'\boldsymbol{\Lambda}_y\mathbf{y}^+$.

In short, an over-dispersed Poisson regression with/without random coefficients is approximated by the following steps: (I) Estimate $\hat{\mu}_i^+$ using a log-Gaussian model whose explained variable $\log(y_i^+) = \log\left(\frac{y_i + 0.5}{z_i}\right) - \frac{1 + 0.5r}{y_i + 0.5}$ and sample weight $y_i + 0.5$; (II) Substitute the estimated $\hat{\lambda}_i^+ = z_i \exp(\hat{\mu}_i^+)$ into Eqs (2)–(4) for models without random coefficients or Eqs (16)—(18) for models with random effects. Later, we examine approximation accuracy of our approach through Monte Carlo experiments.

**Table 1. Closed-form approximations for Poisson regression.** $c$ is a tuning parameter that must be determined a priori. $z_i = 1$ is assumed. All the approximations employ Gaussian linear regression whose explained variables and weights are as shown in the table.

| Method | Explained variables | Weight | Outline |
|---|---|---|---|
| Log-linear approx. [19] | $\log(y_i+c)$ | $y_i+c$ | $\hat{\boldsymbol{\beta}}$ and $Var[\hat{\boldsymbol{\beta}}]$ are estimated from Gaussian model |
| Taylor approx. [20]; Log-Gamma approx. [22] | $\log(y_i + c) - \frac{c}{y_i+c}$ | $y_i+c$ | |
| Our approximation | $\log(y_i + 0.5) - \frac{1+0.5r}{y_i+0.5}$ | $y_i+0.5$ | $\hat{\boldsymbol{\beta}}$ and $Var[\hat{\boldsymbol{\beta}}]$ are estimated from an over-dispersed Poisson regression model whose mean function $\lambda_i$ is approximated using the log-Gaussian model |

## Property of the proposed approximation

Table 1 summarizes closed-form approximations for the Poisson regression models. These methods perform approximations through the estimation of a log-Gaussian model using the explained variables and the weight shown in this table. These practical methods will be useful to avoid the identification problem for not only researchers but also practitioners. However, existing methods are accurate only for a moderate to large $\mu_i$ [22]. These methods should not be used for counts with many zeros. Besides, the $c$ parameter, which has a considerable impact on analysis result, must be determined a priori (see the Introduction section). These drawbacks inhibit the practical use of these approximations.

Major advantages of our approximation relative to these existing methods are as follows: (A) it does not have the tuning parameter ($c$); (B) because of the mode matching, the proposed method accurately approximates the mode of the Poisson distribution irrespective of $\mu_i$; (C) Gaussian approximation is used only for estimating the Poisson mean while alternative methods use it for estimating both the Poisson mean and the regression coefficients. As we will show later, these advantages considerably improve the approximation accuracy for count data with many zeros.

Our mode-matching method is akin to the Laplace approximation, which is based on the mode-matching of a Gaussian distribution and the target distribution. Considering studies demonstrating the accuracy of the Laplace approximation, our mode-based approach is expected to be accurate as well. Still, the Laplace approximation can have poor accuracy if the target distribution is far from Gaussian distribution. Extension based on other approximation methods such as numerical quadrature is an important remining task.

## Results: Monte Carlo experiments

### Case 1: Basic over-dispersed Poisson regression model

This section compares the estimation accuracy of the proposed approximation (Proposed) with standard Poisson regression (Poisson), over-dispersed Poisson regression (odPoisson), and negative binomial regression alternatives (NB). We also compare ours with the log-linear approximation of [19] (LogLinear) and the Taylor approximation of [20] (Taylor).

The simulated count data $y_i$ is generated from the over-dispersed Poisson regression with mean $\lambda_i$ and the overdispersion parameter $\sigma^2$:

$$y_i \sim odPoisson(\lambda_i, \sigma^2), \quad \lambda_i = \exp(\beta_0 + x_{i,1}\beta_1 + x_{i,2}\beta_2), \tag{20}$$

where $x_{i,1}$ and $x_{i,2}$ are generated from standard normal distributions $N(0, 1)$, and $\{\beta_1, \beta_2\} = \{2.0, 0.5\}$. We refer to $\beta_1$ as a strong and $\beta_2$ as a weak coefficient. $\sigma^2 = 1$ implies the standard

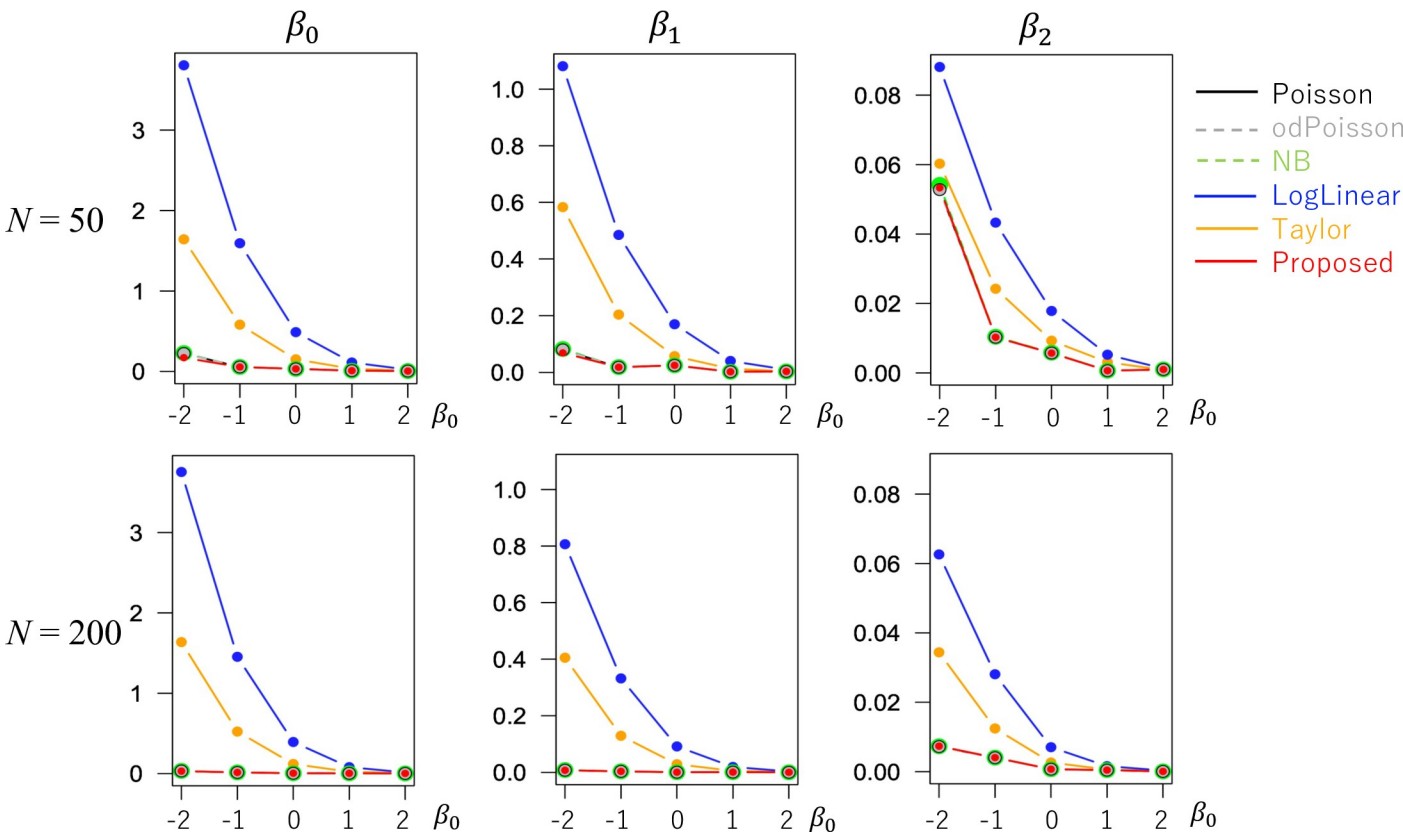

**Fig 1. RMSE of the regression coefficients in cases without overdispersion ($\sigma^2 = 1.0$) (x–axis: $\beta_0$, y-axis: RMSE).**

Poisson regression without over-dispersion while $\sigma^2 > 1$ means over-dispersion. The $\beta_0$ parameter implicitly controls the ratio of zero counts; a smaller $\beta_0$ value yields more zero counts.

Over-dispersed Poisson distribution does not have probability mass function [24]. The simulation data is sampled to satisfy $E[y_i] = \lambda_i$ and $Var[y_i] = \sigma^2\lambda_i$, which $y_i \sim odPoisson(\lambda_i, \sigma^2)$ assumes, as follows:

a. Calculate $\lambda_i = \exp(\beta_0 + x_{i,1}\beta_1 + x_{i,2}\beta_2)$

b. Calculate $v_i = (\sigma^2 - 1)/\lambda_i$

c. Sample $y_i \sim NB(\lambda_i, v_i)$ where $NB(\lambda_i, v_i)$ is a negative binomial distribution with expectation $\lambda_i$ and variance $Var[y_i] = \lambda_i + v_i\lambda_i^2$

The sampled $y_i$ has the expectation $E[y_i] = \lambda_i$ and variance $Var[y_i] = \lambda_i + v_i\lambda_i^2 = \lambda_i + (\sigma^2 - 1)\lambda_i = \sigma^2\lambda_i$ that $odPoisson(\lambda_i, \sigma^2)$ assumes. Thus, the sampled $y_i$ fulfills the assumption of the over-dispersed Poisson distribution.

The coefficient estimation accuracy is compared across models while varying $\beta_0 \in \{-2, -1, 0, 1, 1\}$, $\sigma^2 \in \{1, 5\}$, and $N \in \{50, 200\}$. In each case, the simulations were iterated 1000 times and the root mean squared error (RMSE) and the mean bias are evaluated:

$$RMSE[\beta_k] = \sqrt{\frac{1}{N}\sum\nolimits_{iter=1}^{500}(\hat{\beta}_k^{(iter)} - \beta_k)^2}, \qquad Bias[\beta_k] = \frac{1}{N}\sum_{iter=1}^{500}(\hat{\beta}_k^{(iter)} - \beta_k) \qquad (21)$$

where $\hat{\beta}_k^{(iter)}$ is the estimated $\beta_k$ in the *iter*-th iteration.

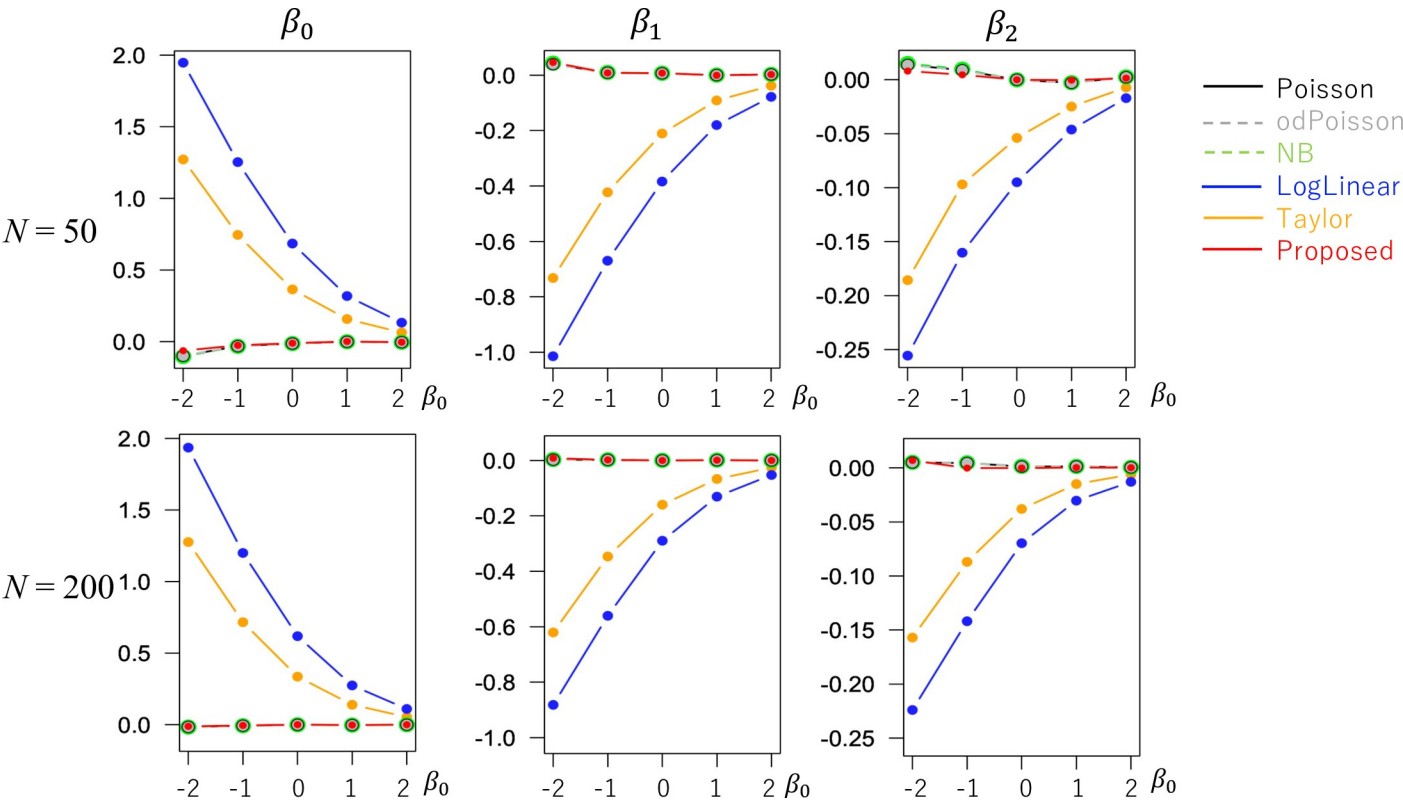

**Fig 2. Bias of the regression coefficients in cases without overdispersion ($\sigma^2 = 1.0$) (x–axis: $\beta_0$, y-axis: Bias).**

The evaluated RMSE and bias values are plotted in Figs 1 and 2 in a case without overdispersion $\sigma^2 = 1.0$ whereas Figs 3 and 4 in cases with overdispersion $\sigma^2 = 5.0$. LogLinear and Taylor tend to have large RMSEs and biases across cases, and the errors inflate if $y_i$ has many zero values (i.e., small $\beta_0$). In contrast, the RMSE values for Proposed are as small as the Poisson and odPoisson specifications across cases. Poisson, odPoisson, and NB have large RMSE values for small over-dispersed samples ($\sigma^2 = 5.0$) with many zero values ($\beta_0 = -2$); it is attributable to the identification problem explained in the "Introduction" section. Proposed does not suffer from this problem. Proposed is advantageous in terms of stability. The bias of the proposed method is small across cases. It is suggested that the proposed method estimates regression coefficients in a reasonable accuracy.

Fig 5 shows the coefficient standard error (SE) estimates. While the SEs estimated from Proposed are similar to odPoisson, the former method tends to have smaller SE values than the latter when $\sigma^2 = 5.0$. To examine if our SE accurately estimates the uncertainty in the coefficient estimates, Fig 6 plots (SE)/(standard deviation of the estimated coefficient values). The value is close to 1.0 if the SEs accurately evaluate the uncertainty. Based on the figure, all the methods tend to underestimate the SE value. Still, the bias of Proposed is smaller than Poisson, NB, LogLinear, and Taylor whereas larger than odPoisson. Reducing the bias in the SE estimates is an important task.

In S1 Appendix in S1 File, we perform another Monte Carlo experiments assuming six explanatory variables. The result is consistent with the results obtained in this section.

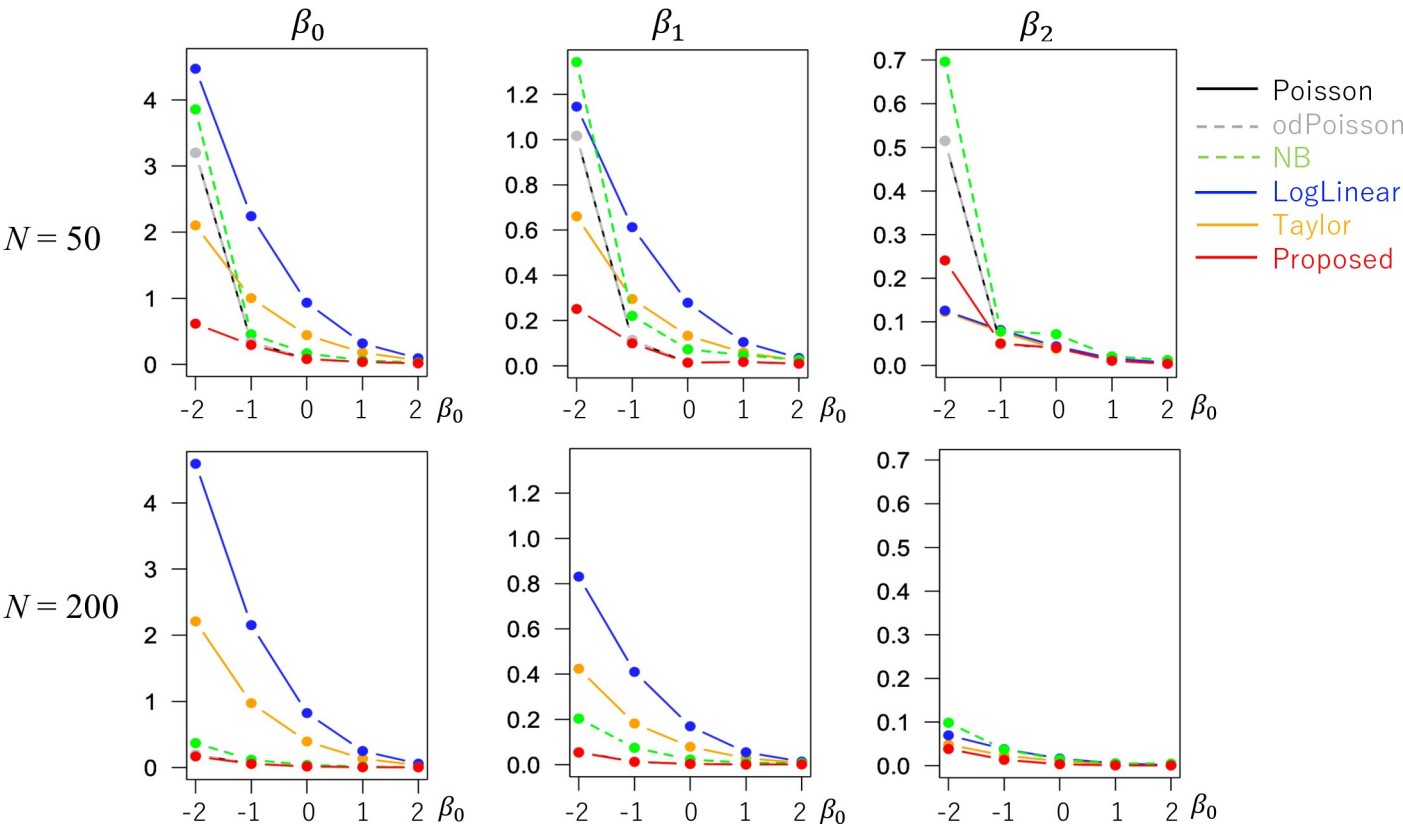

**Fig 3. RMSE of the regression coefficients in cases with overdispersion ($\sigma^2 = 5.0$) (x–axis: $\beta_0$, y-axis: RMSE).**

### Case 2: Model with spatial effects

To verify the expandability of the proposed model, this section applies the proposed method to estimate a spatial regression model, which has been widely used to analyze spatial phenomena in the environment, economy, and epidemic. We consider the following model:

$$y_i \sim odPoisson(\lambda_i, \sigma^2), \quad \lambda_i = \exp(\beta_0 + x_{i,1}\beta_1 + x_{i,2}\beta_2 + s_i), \tag{22}$$

where $\{\beta_1, \beta_2\} = \{2, 0.5\}$ and $\sigma^2 = 5$, which means an overdispersion with variance $Var[y_i] = 5\lambda_i$. $s_i$ is a process capturing a spatially dependent pattern of the data. It is modeled by a low rank Gaussian process whose spatial dependence exponentially decays relative to the Euclidean distance between the geometric centers of the two zones. Eq (22) is an over-dispersed Poisson mixed-effects model (MEM) that considers spatial dependence. The model is estimated by applying the maximum likelihood (ML) estimation for the Poisson MEM (Poisson), an over-dispersed Poisson MEM (odPoisson), the Taylor approximate Poisson MEM (Taylor), and our specification (Proposed). Taylor and Proposed fitted linear MEMs using the transforming explained variables and weight variables (see Table 1). All models were estimated using a restricted maximum likelihood method implemented a R package mgcv [11].

We assumed $\beta_0 \in \{-2, -1, 0, 1, 1\}$ and $N \in \{50, 200\}$. In each case, the models were estimated 500 times, and the estimation accuracies were compared. Figs 7 and 8 display the estimated RMSEs and biases, respectively. When $N = 50$, odPoisson took extremely large RMSEs due to the identification problem. Poisson and Taylor also had large RMSEs. In contrast, the proposed method tends to have smaller RMSE values. The proposed method may be a better

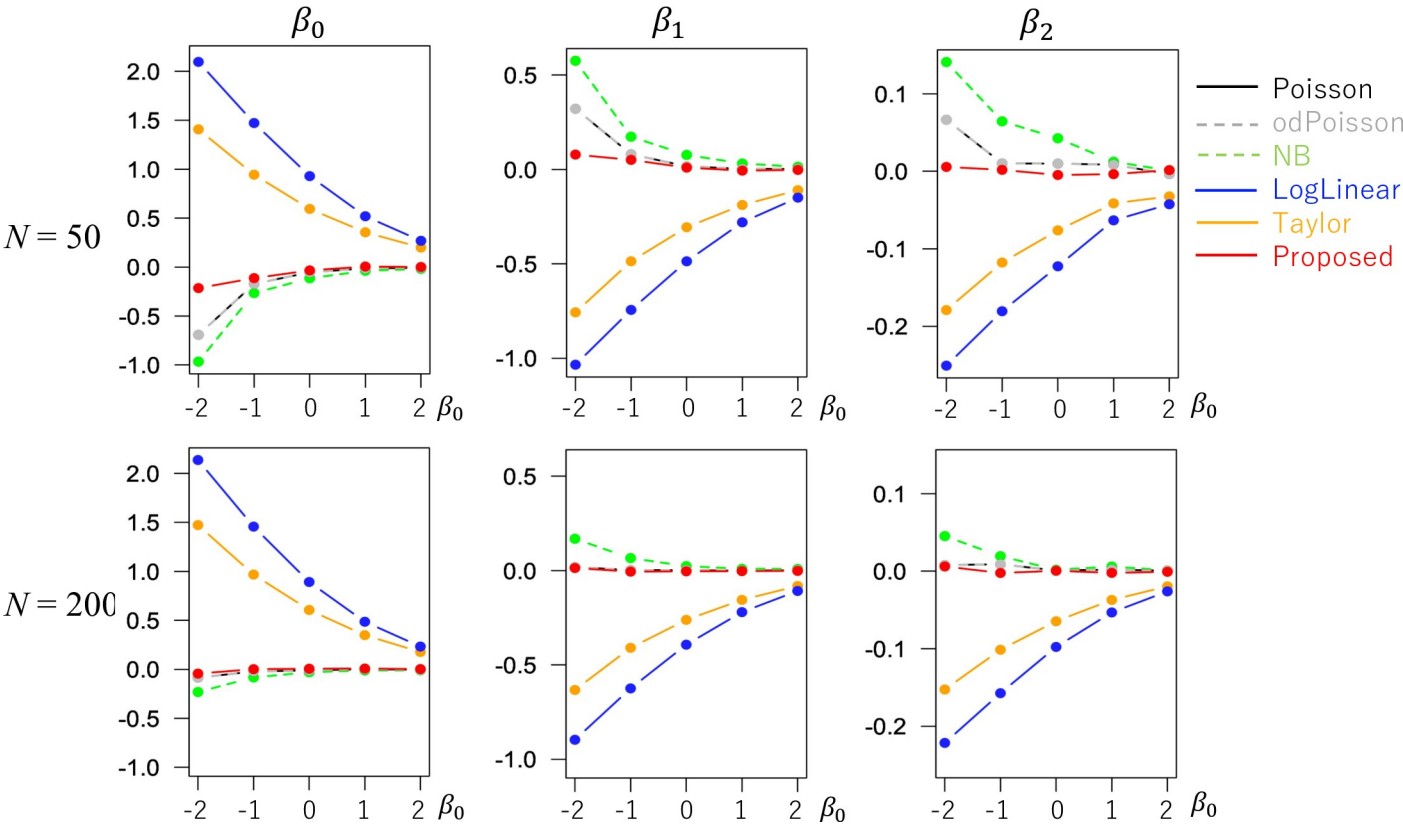

**Fig 4. Bias of the regression coefficients in cases with overdispersion ($\sigma^2$ = 5.0) (x–axis: $\beta_0$, y-axis: Bias).**

choice for small samples. Even for $N$ = 200, the RMSEs and biases of Proposed were as small as those of Poisson and odPoisson. The estimation accuracy of the proposed method was verified in the case of spatial regression.

Fig 9 compares the coefficient standard errors. The SEs obtained from Proposed are similar to odPoisson for a large $\beta_0$, while Proposed has smaller SEs for small $\beta_0$. Fig 10 plots (SE)/(standard deviation of the estimated coefficient values) when $N$ = 200. Unlike odPoisson whose SEs are severely underestimated for small $\beta_0$. Proposed estimates SEs reasonably accurately across cases.

Finally, Fig 11 compares the estimation accuracy for the spatially dependent process $s_i$. The RMSE values of Proposed are almost identical with odPoisson suggesting the accuracy of our approximation.

We performed another Monte Carlo experiment assuming group effects, which estimates heterogeneity across groups, instead of the spatially dependent effects. As summarized in S2 Appendix in S1 File, the RMSEs and biases are as small as Poisson and odPoisson for $N$ = 200 and smaller than the two methods for $N$ = 50.

In short, the proposed method provides an accurate and stable approximation for an overdispersed Poisson MEM.

## Computation time comparison

Finally, computation time is compared while varying $N \in \{1,000, 10,000, 100,000, 300,000\}$ under cases 1 and 2. $\beta_0$ = 0 and $\sigma^2$ = 1 are assumed in this section. We use R version 4.0.2

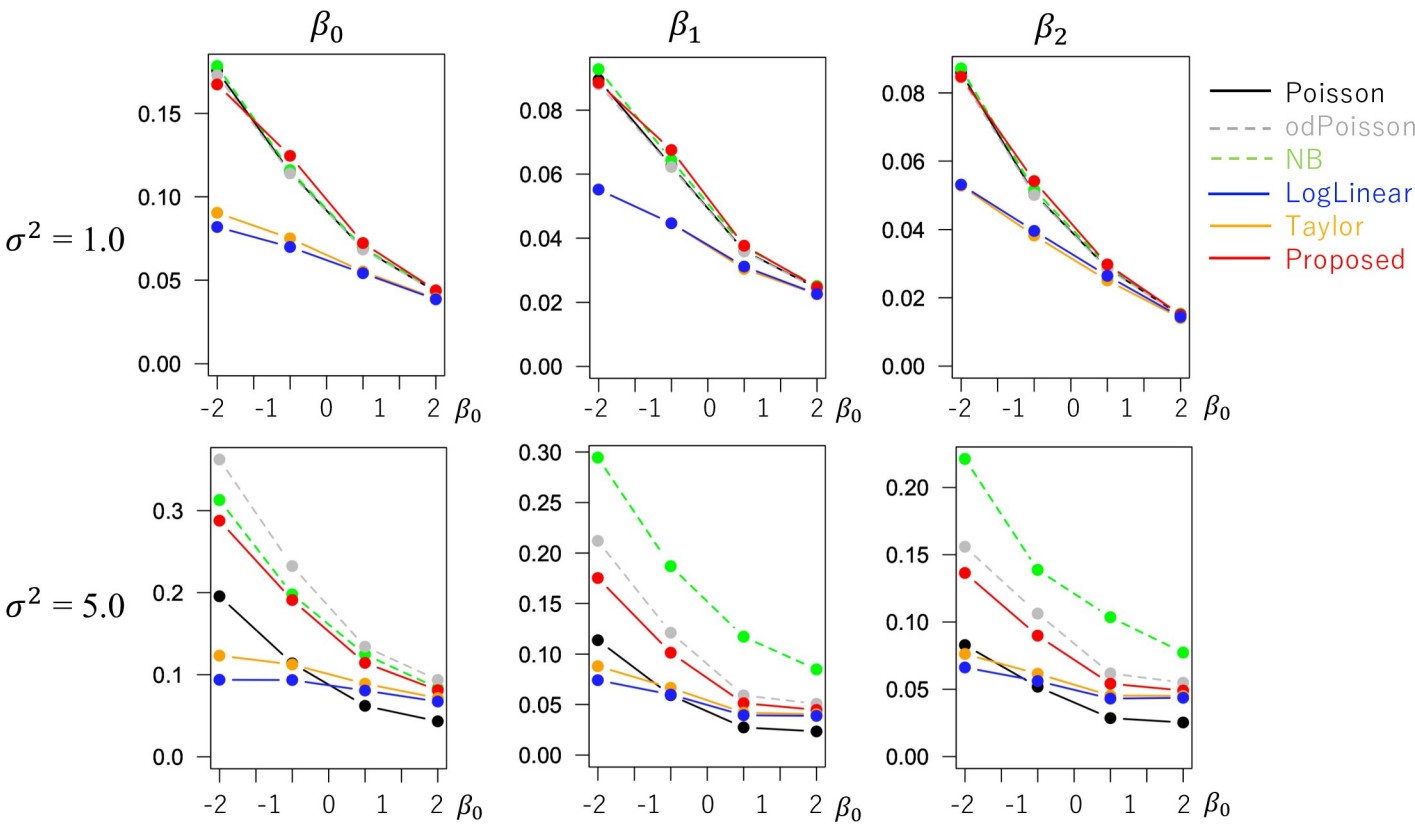

**Fig 5. Means of the coefficient standard errors ($N$ = 200) (x−axis: $\beta_0$, y-axis: mean standard error).**

(https://cran.r-project.org/) installed in a Mac Pro (3.5 GHz, 6-Core Intel Xeon E5 processor with 64 GB memory). The gam function in the mgcv package is used for the model estimation.

Under case 1 (basic model), Poisson, odPoisson, and Proposed took 20.1, 116.0, and 1.34 seconds on average, respectively. In case 2 which estimated spatial effects, Proposed is again

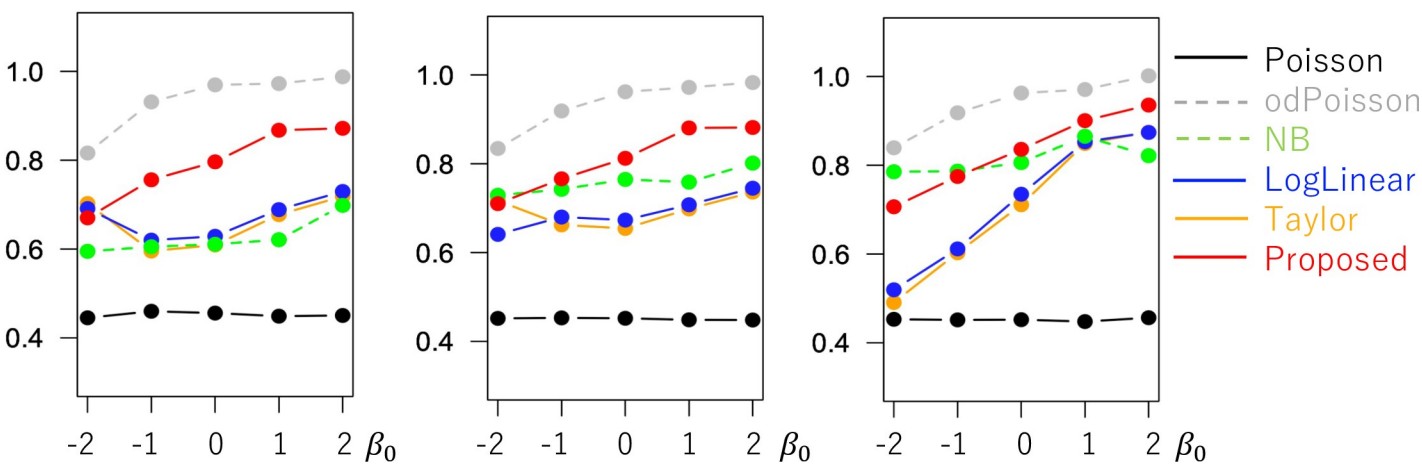

**Fig 6. Means of (estimated standard error)/(standard deviation of the estimated coefficient values) when ($N$ = 200; x−axis: $\beta_0$, y-axis: mean standard error).**

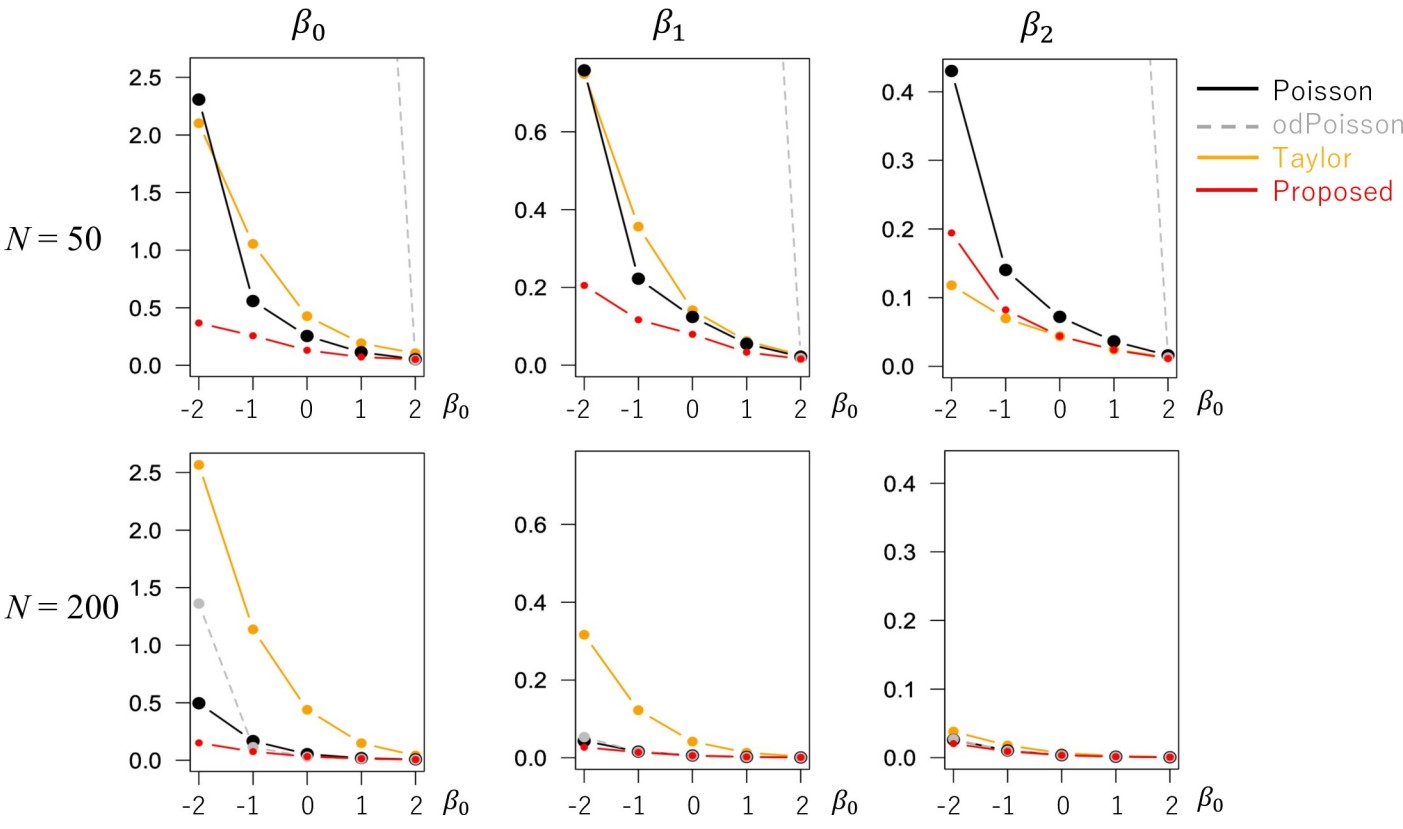

**Fig 7. RMSE of the regression coefficients (model with spatial effects) (x-axis: $\beta_0$, y-axis: RMSE).**

considerably faster than Poisson and odPoisson, especially for large samples as plotted in Fig 12. The computational efficiency of Proposed is confirmed.

## Results: COVID-19 analysis

### Outline

This section employs the developed approximation to an analysis of the COVID-19 (coronavirus disease 2019) pandemic. Since the first case was detected in Wuhan, China, in December 2019, the coronavirus spread. As of February 1, 2021, the cumulative number of confirmed cases is 103.41 million, while the confirmed death toll is 2.25 million. To achieve effective infection control for not only COVID-19 but also pandemics/endemics in the future, it is important to investigate the influencing factor behind the disaster. The data of daily new cases analyzed in this section is provided from JX Press corporation (https://jxpress.net/).

Fig 13 plots the number of daily cases in Japan between February 1, 2020, and January 29, 2021. The number peaked around April 2020, August 2020, and January 2021, respectively. Based on the time trend, we refer to February 1 –May 31 as the first wave, June 1 –September 30 as the second wave, and October 1 –January 29, 2021, as the third wave. Fig 14 displays the spatial plots of the daily new cases by prefecture. This figure shows the tendency of the number of infected people to become large near Tokyo and Osaka, which are major urban areas.

We performed a regression analysis exploring the influencing factor of the increase/ decrease in each wave. The explained variables were the number of daily cases by prefecture by 10-year age groups (-19, 20–29, . . ., 70–79, 80-). The sample sizes were 51,336, 50,508, and

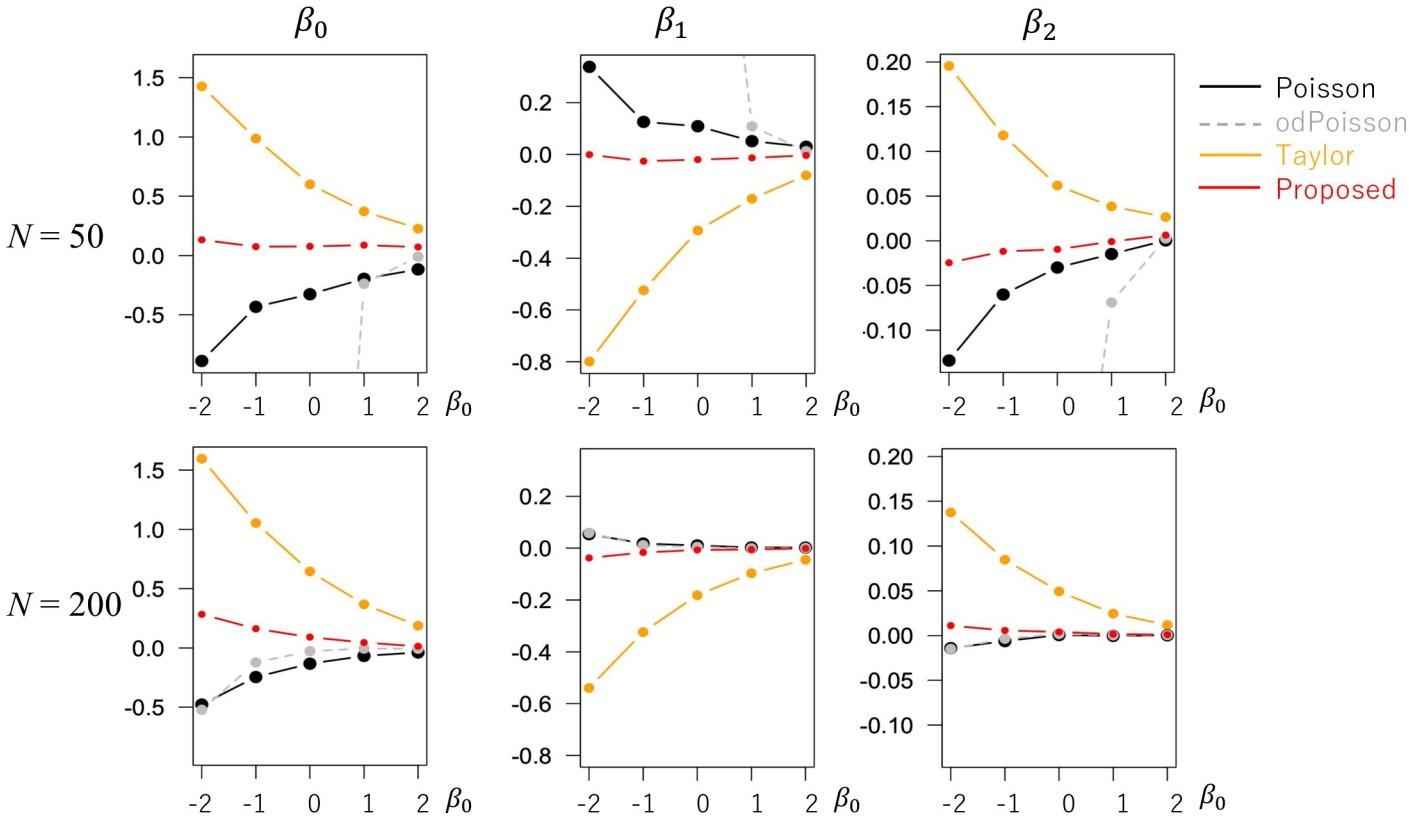

**Fig 8. Bias of the regression coefficients (model with spatial effects) (x-axis: $\beta_0$, y-axis: Bias).**

50,094 for the three waves, respectively. 89.0% (45,696 samples), 77.1% (38,954 samples), and 49.7% (24,873 samples) of the samples were zeros.

For the COVID-19 data, we approximate the following over-dispersed Poisson additive mixed model using our approach:

$$y_i \sim odPoisson(\lambda_i, \sigma^2), \quad \lambda_i = \exp(\beta_0 + x_i\beta_1 + \sum_{l=1}^{4} g_{i,l} + s_i), \tag{23}$$

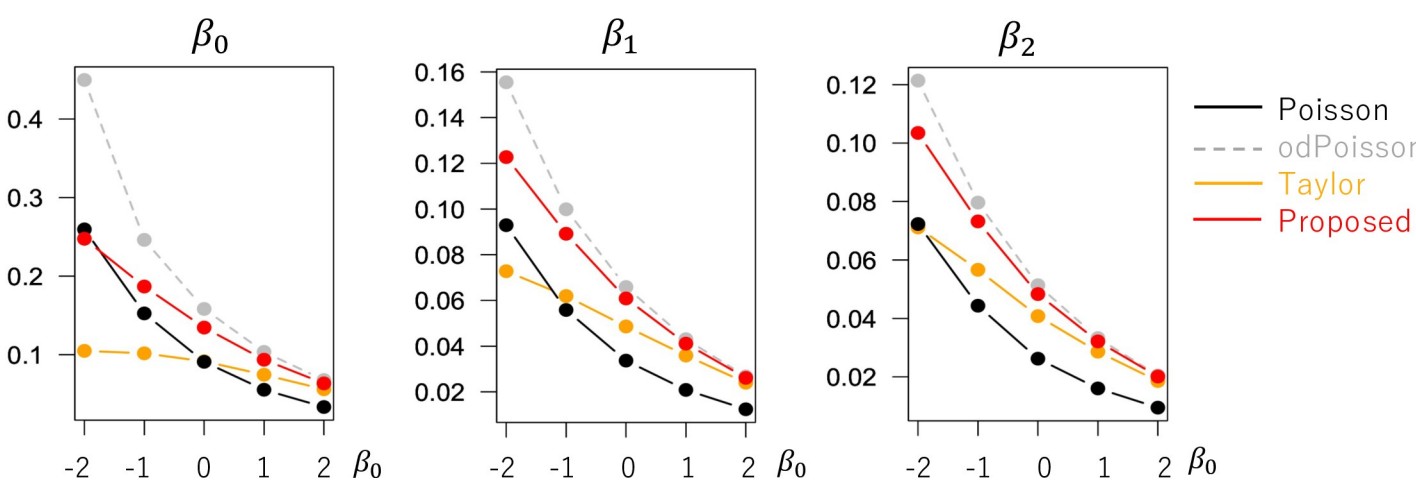

**Fig 9. Means of the coefficient standard errors ($N = 200$) (x–axis: $\beta_0$, y-axis: mean standard error).**

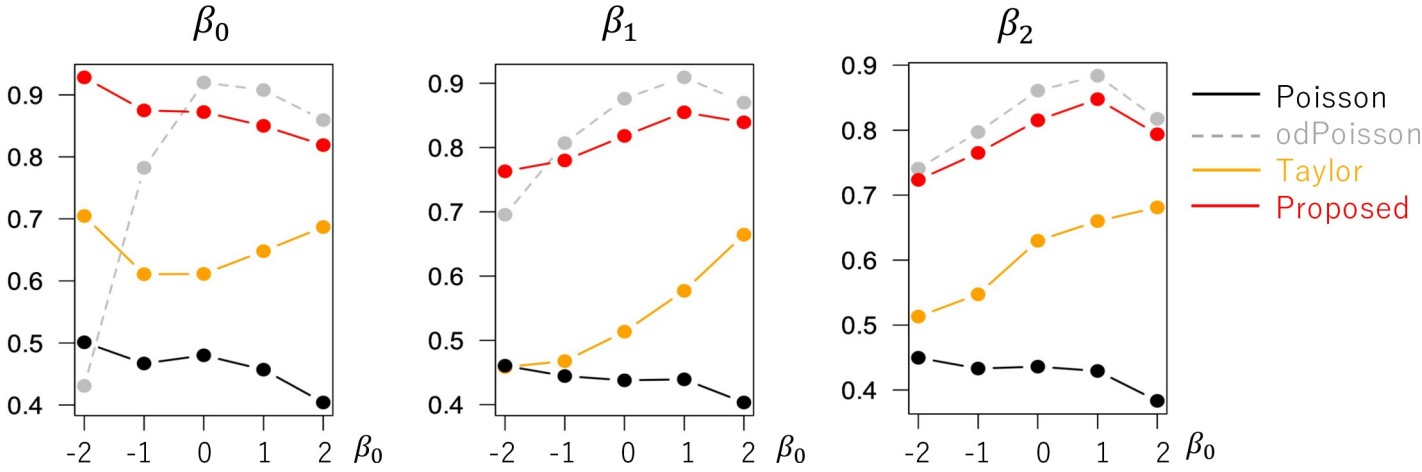

**Fig 10. Means of (estimated standard error)/(standard deviation of the estimated coefficient values)** ($N = 200$; x–axis: $\beta_0$, y-axis: RMSE).

where $y_i$ is the number of daily new cases. $\beta_0$ and $\beta_1$ are regression coefficients. The model is fitted for each of the three datasets completely separately. To scale the mean function according to the population, the offset variable $z_i$ is given by the prefectural population. The explanatory variable $x_i$ is the prefectural pedestrian density by day, which is relative to January 13, 2020 (source: Apple Mobility Trends: https://covid19.apple.com/mobility). The density is estimated based on the number of route searches by Apple map users. For further detail, see the

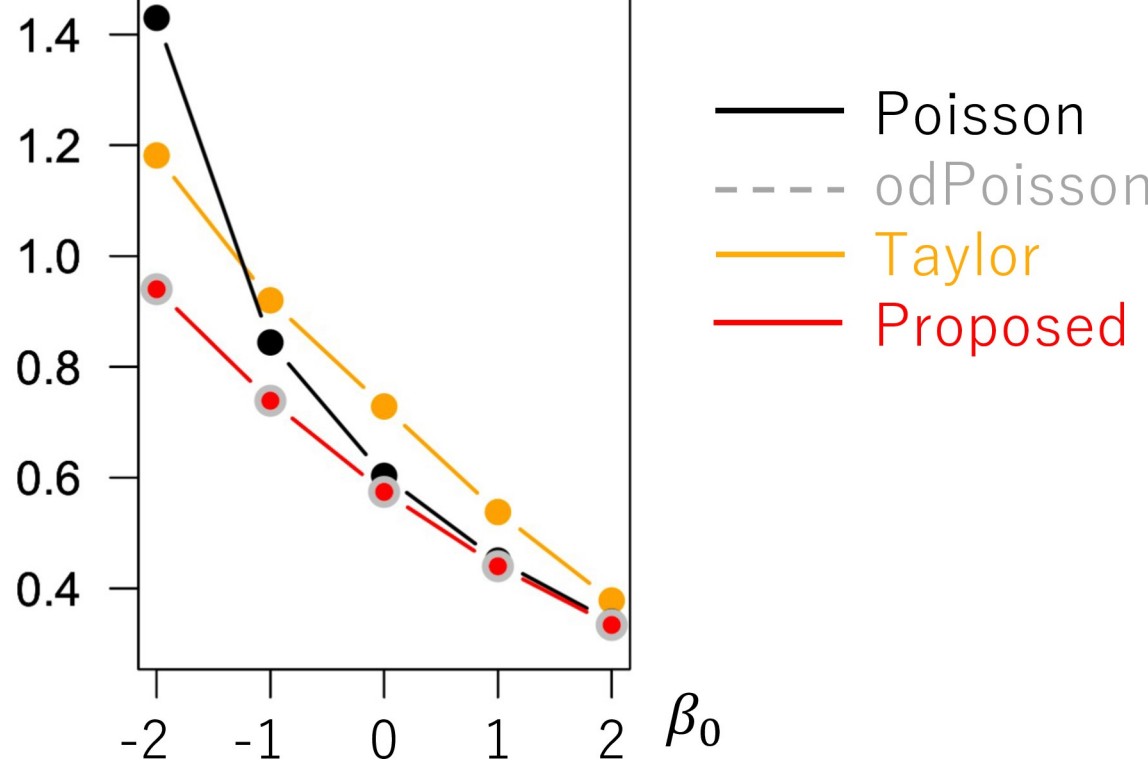

**Fig 11. RMSE of the estimated spatial effects** (x-axis: $\beta_0$, y-axis: RMSE).

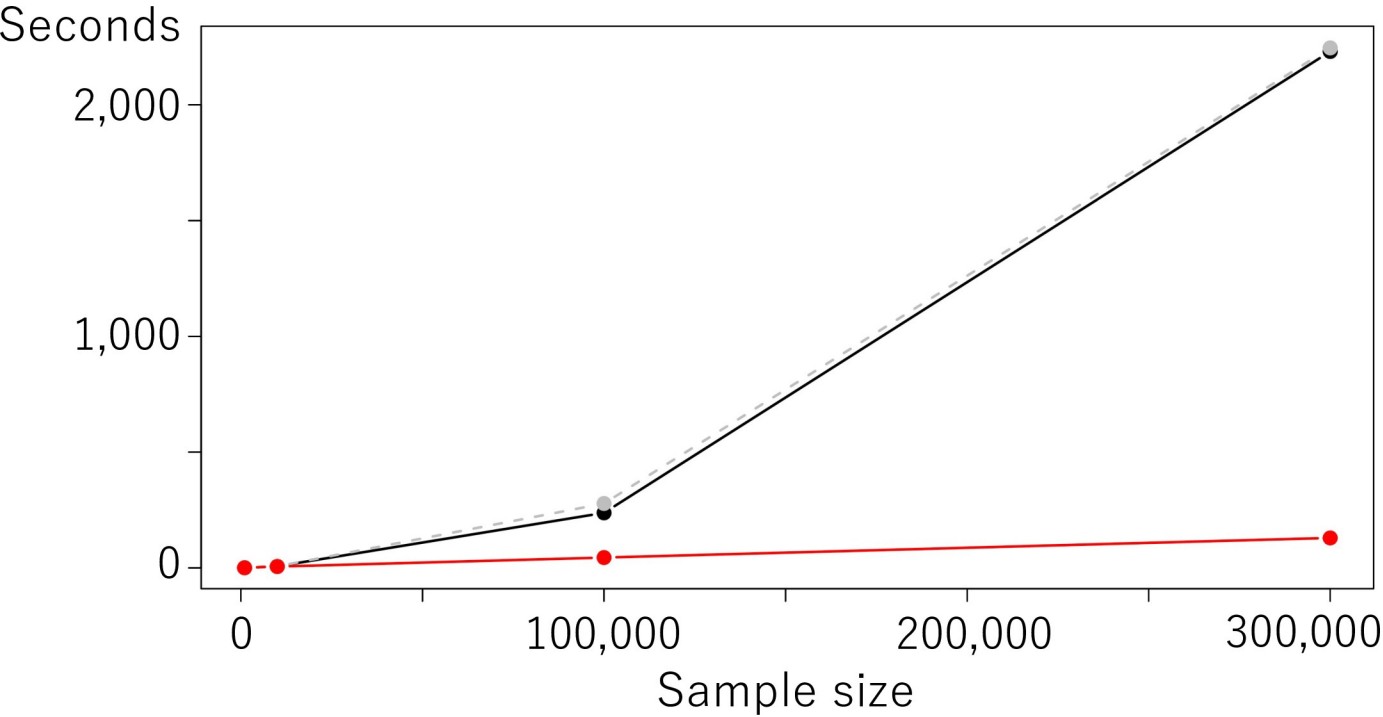

**Fig 12. Computation time comparison under case 2 (model with spatial effects).**

source page. $g_{i,l}$ represents the $l$-th group-wise random effect. We consider the effects by week ($g_{i,1}$), days of the week ($g_{i,2}$), generation ($g_{i,3}$), and prefecture ($g_{i,4}$). In considering countermeasures, it is important to reveal not only patterns by prefectures but also across prefectures. To estimate this, we include a low rank Gaussian process $s_i$ which smoothly varies depending on geographic coordinates; we use the geographic center of each prefecture for the modeling. The model was estimated using the mgcv package.

## Results

Table 2 summarizes the estimated parameters. The estimated coefficients of pedestrian density become positively significant in the first and second the waves. Self-restraint was estimated to reduce the number of cases in the early periods. Based on the estimated dispersion parameter ($\sigma^2$), the variance of the number of cases were over-dispersed, and the tendency became stronger over time.

Fig 16 plots the estimated group effects by week, days of the week, and generation. The estimated week-wise effects show that the increase in cases lasts longer in the third wave. Control of the infection spread might be getting more difficult over waves. Regarding the days of the week, Monday has the lowest while Thursday, Friday, and Saturday have higher values. The difference is attributable to some business reasons such as the closing of hospitals and PCR test sites. The estimated generation effects have considerable differences across waves. In the first wave, people who are in the working generation (the 20s - 50s) tend to be infected. Commuting and/or meeting in the office might trigger the infection. In the second wave, the 20's group has a strong tendency of being infected as compared to the elders, therefore, more self-restriction is needed. In the third wave, not only the 20s but also the 30s – 50s have high chances of being infected. Infection might spread again across the working generation.

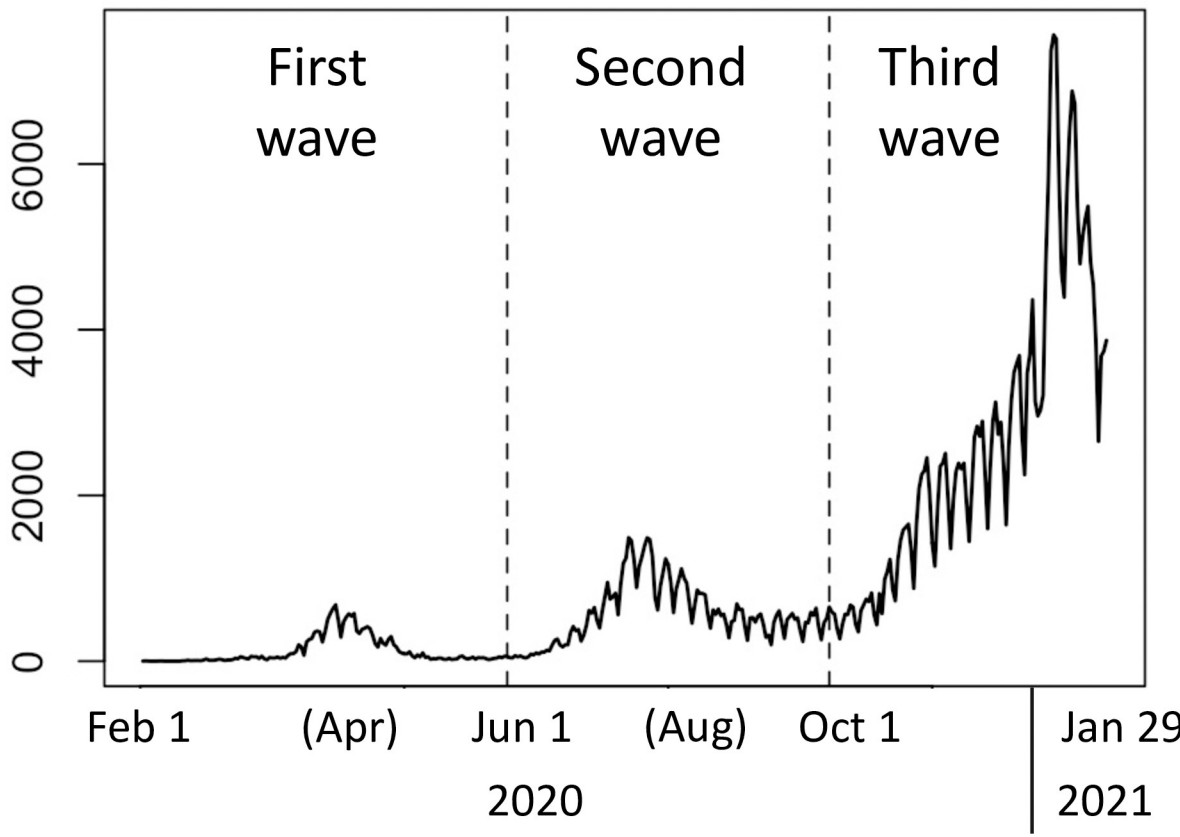

**Fig 13. Daily number of cases across Japan.**

Fig 17 plots the estimated prefecture-wise independent effects and spatially dependent effects. The former estimates local hotspots while the latter, global hotspots. The estimated prefecture-wise effects suggest that prefectures including major cities (Tokyo, Osaka, Fukuoka) and Hokkaido are local hotspots. More countermeasures might be required in these prefectures. On the other hand, based on the estimated spatially dependent effects, there is a global hotspot around Tokyo, and the influences grow over waves. Control of the infection spread from Tokyo might have been important to mitigate the third wave.

## Discussion

This study develops a practical log-Gaussian approximation for Poisson regression models. Considering its simplicity, stability, and computational efficiency, it will be useful for researchers as well as practitioners.

Exploring the expandability of our approach is an important future task. For example, our approach might be useful for spatial and spatiotemporal interpolation of count data by combining it with Gaussian process models without additional computation and implementation costs. Our approach might also be useful for fast count data assimilation by combining it with a state-space model. Exploring such extensions will be an interesting research endeavor.

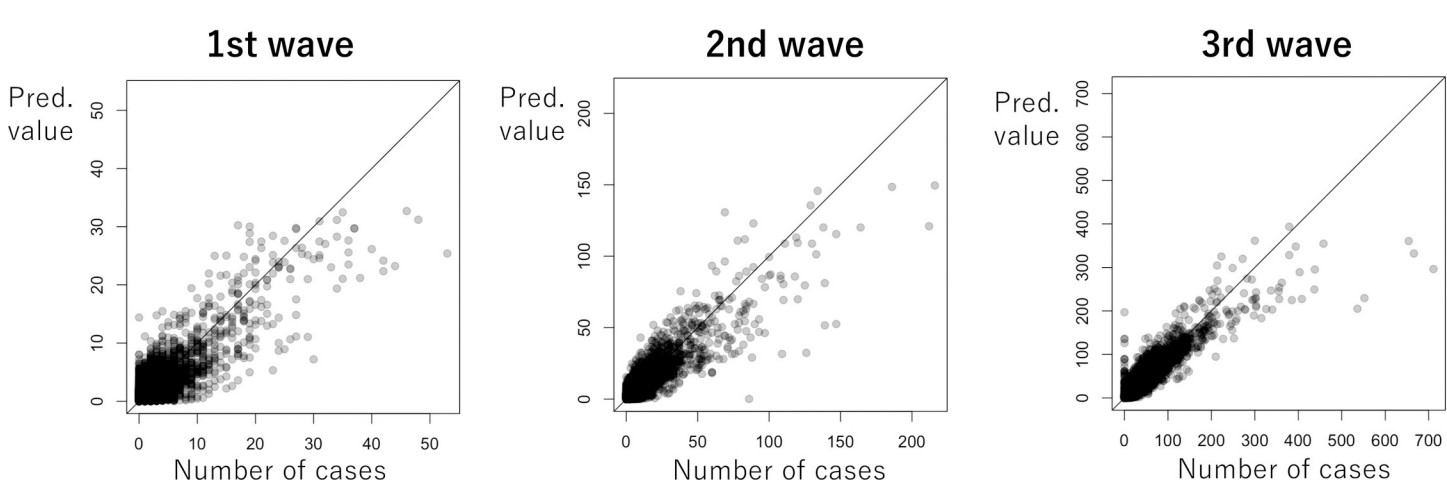

**Fig 14. Number of cases by prefecture.**

**Fig 15. Comparison of the observed and predicted number of cases.**

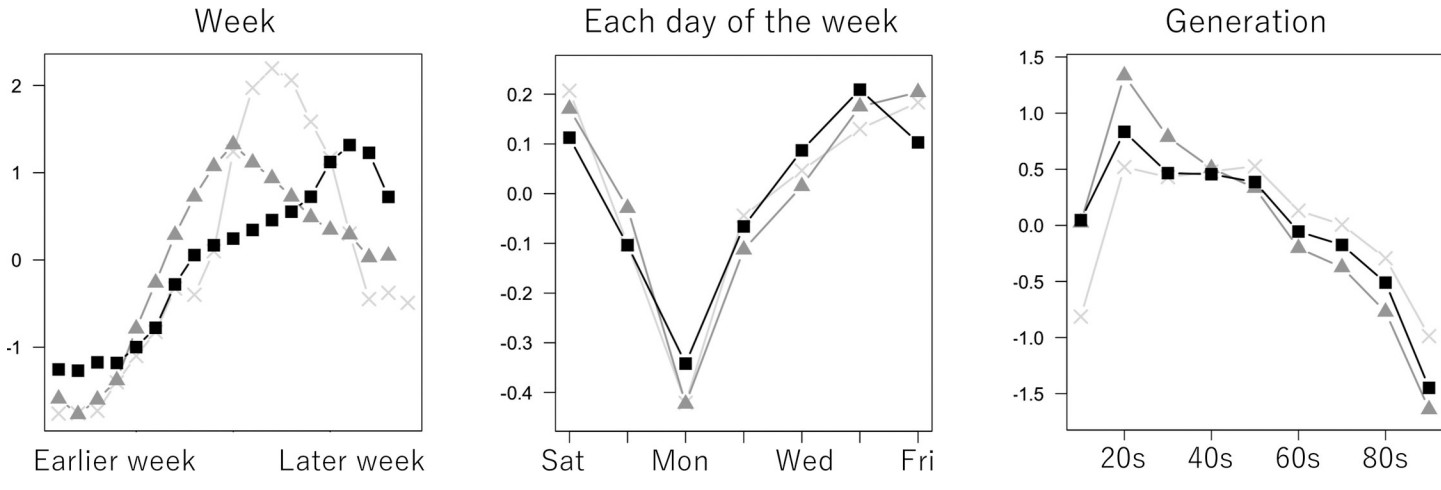

**Fig 16. Estimated group effects (week, days of the week, generation).**

**Fig 17.** Estimated group effects by prefecture (top) and spatially dependent effects (bottom).

**Table 2. Parameter estimates.** See Fig 15 for the fitting on the number of cases.

| | First wave | | Second wave | | Third wave | |
|---|---|---|---|---|---|---|
| | Est. | t value | Est. | t value | Est. | t value |
| Const ($\beta_0$) | −18.03 | −84.71 ***[1] | −17.91 | −84.13 *** | −14.64 | −68.79 *** |
| Pedestrian density ($\beta_1$) | 0.31 | 4.26 *** | 1.30 | 18.01 *** | 0.09 | 1.29 |
| Dispersion parameter ($\sigma^2$) | 1.29 | | 2.22 | | 3.33 | |

[1] *** demotes the statistical significance of 1%.

## Supporting information

**S1 File.**
(DOCX)

## Author Contributions

**Conceptualization:** Daisuke Murakami.

**Data curation:** Tomoko Matsui.

**Formal analysis:** Daisuke Murakami.

**Funding acquisition:** Daisuke Murakami.

**Methodology:** Daisuke Murakami.

**Writing – original draft:** Daisuke Murakami, Tomoko Matsui.

**Writing – review & editing:** Tomoko Matsui.

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
