## [Decision Letter · Decision Letter 0]

26 May 2021

PONE-D-21-12939

Improved log-Gaussian approximation for over-dispersed Poisson regression: application to spatial analysis of COVID-19

PLOS ONE

Dear Dr. Murakami,

Thank you for submitting your manuscript to PLOS ONE. After careful consideration, we feel that it has merit but does not fully meet PLOS ONE’s publication criteria as it currently stands. Therefore, we invite you to submit a revised version of the manuscript that addresses the points raised during the review process.

In particular:

Improve clarity of the manuscriptClarify why the reported main motivation is improved efficiency and accuracy for Bayesian Poisson regression but Bayesian modelling does not appear anywhere in the paperClarify the meaning of "posterior variance" etc.Explain and motivate mismatch between the model used to simulate data and the model used to analyse itClarify the reason for the “extra” 0.5/(Y+0.5) in (6)Revise terminology (e.g. zero-inflated Poisson, posterior variance, etc)Poisson regression with random effects/spatial effects is already very well developed, so approximations such as this one are not strictly necessary. Please demonstrate that the proposed solution is much more efficient.

We look forward to receiving your revised manuscript.

Kind regards,

Luca Citi, PhD

Academic Editor

PLOS ONE

Journal Requirements:

2. Please include the data sources used in the Data availability statement and Methods section. We note that the source of the COVID-19 data appears to be unclear in the Methods section. Please also indicate in the Data availability statement whether you are able to openly share the code used, and if so, where others can access this.

3.We note that Figure(s) 10 and 13 in your submission contain map images which may be copyrighted. All PLOS content is published under the Creative Commons Attribution License (CC BY 4.0), which means that the manuscript, images, and Supporting Information files will be freely available online, and any third party is permitted to access, download, copy, distribute, and use these materials in any way, even commercially, with proper attribution. For these reasons, we cannot publish previously copyrighted maps or satellite images created using proprietary data, such as Google software (Google Maps, Street View, and Earth). For more information, see our copyright guidelines: http://journals.plos.org/plosone/s/licenses-and-copyright.

a) You may seek permission from the original copyright holder of Figure(s) 10 and 13 to publish the content specifically under the CC BY 4.0 license. 

Additional Editor Comments:

Please clarify the meaning of, e.g., "posterior variance". In a Bayesian setting "prior" and "posterior" usually refer to distributions of the parameters. So when talking about a Poisson distribution, the "posterior" would be the a distribution of lambda (given one or more observations y), not of the Poisson RV "y" as apparently used in the paper. (See for example https://stats.stackexchange.com/a/26225).

Reviewers' comments:

Reviewer's Responses to Questions

**Comments to the Author**

1. Is the manuscript technically sound, and do the data support the conclusions?

Reviewer #1: Yes

Reviewer #2: No

Reviewer #3: Partly

2. Has the statistical analysis been performed appropriately and rigorously? 

Reviewer #1: Yes

Reviewer #2: No

Reviewer #3: Yes

3. Have the authors made all data underlying the findings in their manuscript fully available?

Reviewer #1: Yes

Reviewer #2: No

Reviewer #3: Yes

4. Is the manuscript presented in an intelligible fashion and written in standard English?

Reviewer #1: Yes

Reviewer #2: No

Reviewer #3: Yes

5. Review Comments to the Author

Reviewer #1: The authors need to used more than 2 explanatory variables. For example, p=4 and 8. ............................................................................................................................................................................................................................................................................................................................................................................................................................................................................

Reviewer #2: You propose an approximation for modeling Poisson or overdispersed (OD) Poisson data, evaluate some of its properties by simulation, and apply it to a data set on prefecture-level data on new Covid-19 cases. Modern computing has made it possible to model generalized linear mixed models, including models with spatial and temporal correlation. So approximations such as yours are not strictly necessary. You suggest that your approximation is a more computationally efficient. I agree this is likely, but you never demonstrate how much more efficient.

I found it difficult to follow some parts of your presentation because of word choice and other issues, perhaps related to translation into English. As a result some of my concerns may simply reflect an unclear presentation. However, I have three major concerns about your approach

I don’t follow where the “extra” 0.5/(Y+0.5) in (6) comes from (lines 92-93). If Z ~ logN(m, v), by definition log Z ~ N(m,v). L 92 looks like you are computing E Z = exp(m + v/2) so the “extra” 0.5/(y+0.5) is v/2. But, the expectation in L 92 is not the equivalent of E Z, it is the equivalent of E log(Z), which = m. Said another way, It looks like you are computing log E Z, which does not equal E log Z. You want E log Z to get to equation (6). The choice of constant in log(Y+c) does matter. One example, where X is transformed, is Ekwaru and Veugelers, 2018, Stat BioPharm Res, 10:26-29.

I also could not follow or do not agree with your variance manipulations. This has two aspects.

1) Line 79 gives, according to your text, an approximation of the “posterior variance of a Poisson distribution”, citing El-Sayyad 1973. I did not read El-Sayyad and do not understand what you mean by “posterior”. Is 1/(y+c) is an approximate variance for a Poisson random deviate or an approximate variance for a log transformed Poisson random deviate?

2) There appears to be a mismatch between your OD variance model and the mechanism used to simulate OD Poisson data. It was not clear in the main text how simulated from an OD Poisson distribution. I finally found the answer in Appendix S1, where you simulate Y ~ Pois(lambda), log lambda = mu + Z, Z ~ N(0, sigma^2). That generates observations from a log normal Poisson mixture for which variance Y = mu + (exp(sigma^2)-1) mu^2. You give your OD variance model as Var log Y = sigma^2/(y+c) approximately. That is not the same variance model as the variance pattern of your simulated data. Applying the Jacobian of the log Y transformation to the log normal variance, you get Var log Y approx. = 1/mu + (exp(sigma^2) -1). I presume El-Sayyad then uses a plug-in estimate of mu to estimate 1/mu as 1/(y+c). These two expressions are the same for Poisson data, i.e. when sigma^2 = 0, but not the same for OD Poisson data.

You emphasize the computational efficiency. Please provide data on that. Compare the time to fit a model using a log-Gaussian approximation to the time required to fit a Poisson or OD Poisson model. The OD Poisson model could be fit by adding an observation specific random effect (to generate a log-normal Poisson distribution).

Details:

Please give this a careful read for English usage and word choice. Three examples are line 33, I presume you meant “collected” not “corrected”, the usual interpretation of “zero-inflated”, and L 226, a “decade” is ten years. There are others throughout the manuscript.

L 45. I don’t understand the “lack of contiguity”. In the usual setup for Poisson count data with latent variables, the Poisson distribution is the data model conditional on the latent observation-specific mean. That latent variable has support on the non-negative half line, so a log-normal or gamma distribution for the latent variable is completely compatible. I believe you’re thinking about approximating a discrete Poisson distribution by a continuous distribution, which has nothing to do with Bayesian Poisson regression.

L 75: Do you mean “prior” here or are you really talking about random effects?

L 99, 108: “sample weight” has two contrasting uses: Variance = weight * constant and weighted SS = Sum weight * (y - \\hat Y)^2. The weighted SS use is more frequent, because that’s how weights are specified in most (all?) software. You’re using the first definition. I suggest you explain carefully, avoiding the term weight, if you retain 1/(y+0.5). Or, define the weight as y+0.5.

L 101: If the “extra” 0.5/(y+0.5) in (6) is correct and your variance model is assumed corrected, notice that adding overdispersion will change the “extra” term to 0.5 \\sigma^2/(y+0.5). The weights won’t change because they are unaffected by a constant multiplier to the variance.

Table 1. It would be useful to add your overdispersed approximation to this table.

L 142. 500 simulated data sets seems rather small to provide a reasonably precise estimate of the rMSE, especially when it seems some estimates are “wild”. Please either provide an estimate of the Monte-Carlo variance to demonstrate that your results from 500 simulations are sufficiently precise or increase the number of simulations (1000, or substantially more if the distribution of estimates is very skewed).

L 148: Please be careful about your terminology. Equation (9) is not generating zero-inflated Poisson data. It is generating data from a Poisson distribution with mean close to 0 so there is a high probability of observing 0. A zero-inflated Poisson model is a mixture model. One mixture component is a point mass at 0, with a mixture probability of p0. The second mixture component is a draw from a Poisson distribution, with mixture probability of 1-p0. Similar issue on l 227. The counts may be zero inflated, but reporting the %0’s doesn’t tell you that.

L 181. Please cite one of the papers describing mgcv instead of linking to the CRAN page. See the information provided by citation(‘mgcv’).

L 192. Small se’s are good only when they still correctly describe the uncertainty in the estimate. One way to check this is to compare the average estimated variance of a coefficient = average squared standard error to the observed variance of an estimate over simulations.

L 221/2, Figures 10 and 11. The figure numbering seems reversed from what is described in the text.

L 225: “generation” – please be consistent in your choice of names. I believe this is what was called wave in lines 216-217. Ahh (based on text later on): generation is age group.

L 225: Please emphasize that you are doing completely separate analyses for each wave of data. One could look at “determinants within each wave” by fitting a single model with a wave*determinant interaction.

L 225: determinants (plural) seems an overstatement. Your model only looked at pedestrian density.

L 226: This is spatio-temporal data. Please explain carefully what data were used in the analysis. The text says “every decade”. Was this one day’s data, the sum over a series of days within “the decade”, all the daily observations within “the decade”, or something else?

L 237 – 239: The spatial component of the data is per prefecture. Why include both prefecture and the spatial random effect in the model? Prefecture will capture the only spatial aspect of the data, so I don’t see where there is residual spatial dependence.

L 240: Why use the spmoran package instead of gamm() in mgcv? GAMM is the spatial method you evaluated in the simulation. If you need to use Moran Eigenvector Maps to model the spatial dependence, please explain the reason.

Reviewer #3: The authors have carried out some commendable work on approximating the Poisson likelihood using a mode-matching log-Gaussian approach. However, the main motivation -- as mentioned in the abstract and introduction -- is on improved efficiency and accuracy for Bayesian Poisson regression, but Bayesian modelling does not appear anywhere in the paper. In all the Results section (including two simulations and the Japanese COVID-19 data analysis), the proposed approximation is only used for frequentist modelling via maximum likelihood and no Bayesian analysis is considered. Given that frequentist estimation and inference for Poisson regressionb with random effects/spatial effects is already very well developed via the MGCV and LME4 packages in R, the advancement using proposed approximation is minimised.

Specific comments include:

Page 2 line 33: "corrected" = "collected"

Page 2 line 45: "contiguity" should be "conjugacy"?

Page 3 Equation (2): the observation y appears on both sides of expression (2). It is mentioned in the following paragraph that "1/(y+c) approximates the posterior variance of the Poisson distribution" but what is this the posterior variance of? Is it the variance of the posterior distribution of P(lambda|y)?

Page 4 Equation (8): the expression here is not limited to over-dispersion. Some discussion (or simulations) based on under-dispersed counts would be interesting in this paper.

Page 6 line 135: how is the over-dispersed Poisson simulated in your paper? Did you use the Neg-Bin, or Poisson-Gamma, or some other simulation mechanism?

Page 6 Equations (10): Would it be better to replace the RMSE by a relative RMSE, by dividing the squared difference by beta_k?

All the figures are very low resolution: I would recommend using a different image format for your figures.

6. PLOS authors have the option to publish the peer review history of their article (what does this mean?). If published, this will include your full peer review and any attached files.

Reviewer #1: No

Reviewer #2: No

Reviewer #3: No

---

## [Author Response · Author response to Decision Letter 0]

15 Jul 2021

Our responses are summarized in an submitted matarial.

---

## [Decision Letter · Decision Letter 1]

6 Sep 2021

PONE-D-21-12939R1Improved log-Gaussian approximation for over-dispersed Poisson regression: application to spatial analysis of COVID-19PLOS ONE

Dear Dr. Murakami,

Thank you for submitting your manuscript to PLOS ONE. After careful consideration, we feel that it has merit but does not fully meet PLOS ONE’s publication criteria as it currently stands. Therefore, we invite you to submit a revised version of the manuscript that addresses the points raised during the review process.

In particular, there are a number of important outstanding issues raised by reviewer 2 about the derivation that should be addressed in the next submission.

We look forward to receiving your revised manuscript.

Kind regards,

Luca Citi, PhD

Academic Editor

PLOS ONE

Journal Requirements:

Additional Editor Comments (if provided):

Reviewers' comments:

Reviewer's Responses to Questions

**Comments to the Author**

1. If the authors have adequately addressed your comments raised in a previous round of review and you feel that this manuscript is now acceptable for publication, you may indicate that here to bypass the “Comments to the Author” section, enter your conflict of interest statement in the “Confidential to Editor” section, and submit your "Accept" recommendation.

Reviewer #1: All comments have been addressed

Reviewer #2: (No Response)

2. Is the manuscript technically sound, and do the data support the conclusions?

Reviewer #1: Yes

Reviewer #2: Partly

3. Has the statistical analysis been performed appropriately and rigorously? 

Reviewer #1: Yes

Reviewer #2: No

4. Have the authors made all data underlying the findings in their manuscript fully available?

Reviewer #1: Yes

Reviewer #2: No

5. Is the manuscript presented in an intelligible fashion and written in standard English?

Reviewer #1: Yes

Reviewer #2: No

6. Review Comments to the Author

Reviewer #1: The authors have adequately addressed all the comments that raised in a previous round of review..............................................................................

Reviewer #2: My comments focus on the adequacy of your response to three issues raised in the first review. You have made an appropriate and major response to two issues, appear to misrepresent a third, and have ignored the fourth. Specifically:

Thank you for the major revision to lines 74 – 139. That makes it much easier to understand your derivation. Unfortunately, it also makes it possible to identify places where the derivation is not done carefully enough. Further comments on this are in the details.

Thank you also for including computing time information. That is helpful.

I remain very concerned about the mismatch between the distributions assumed in the analysis and those used to simulate data. You cite Gsteiger et al to support using a negative binomial distribution to simulate data from the quasi-poisson distribution (i.e. where variance = lambda_i sigma^2). Where does Gsteiger say this? The text below equation (10) in section 2.2 very clearly says variance = mu + k mu^2, the usual expression for negative binomial variance. As I said in the first review, these are two very different variance patterns.

I repeat my previous concern about “too small” SEs. A small SE is a good only when that se is correctly estimated. You claim you can’t evaluate the performance of the SE estimator. I don’t understand why not. If it is numerical issues, doesn’t this raise concerns about the practical use of your approximation.

Specific details:

Equation (2) doesn’t make probabilistic sense because y_i occurs in the specification of the distribution. Technically, you are giving the distribution of y_i conditional on y_i, which is meaningless. Similar issues occur through (6). A derivation could be done more carefully by using E y in the variance expression then substituting y_i as a “plug-in” estimator of E y_i at the end. That will clarify what is being assumed by the proposed approximation. Many do not like plug-in estimators, so you are free to find another way to derive your results that avoids a distribution of Y conditional on Y.

Please check the derivation of equation (7). You seem to be using Var y^* = exp(mu + v), not exp(mu+v/2). The issue of y_i occuring in the variance expression is even more important here.

L 125. Don’t you apply (6) with probability 1-r and (8) with probability r, where r is the proportion of 0 counts?

L 128: lines 125-126 suggest a finite mixture of multiplicative constants, i.e. that from (6) with probability (1-r) and that from (8) with probability r. That does not lead to the definition of y^+ given here. This definition appears out of thin air. It does match (6) when r=0 and (8) when r=1, but that is neither a probabilistic or statistical justification for that definition.

L 135: what happened to r in the definition of the response variable? This is probably a typo because the analogous expression in table 1 includes r.

L 157: Yes, Laplace is often good but it doesn’t always work. There is a substantial literature on better approximations, e.g. numerical quadrature with more quadrature points.

L 203: Please be careful not to overstate your results. Estimates may be close on average, but that does imply they are accurate. I am especially concerned about B0, which seems (Figure 2) just as biased for N=200 as it is for N=50. If you consider an even simpler situation with a constant mean, inspection of equation (6) shows that estimates of B0 are not unbiased and not consistent, because of the approximation.

L 210: What is sigma^2 here? I.e., what distribution is that a parameter of? If a negative binomial distribution is used to simulate data, I strongly suspect that data simulated with sigma^2 = 1 are still overdispersed, based on second major comment about variance models.

L 315: overall, age-group is much clearer than generation. Thank for changing that wording. Here is one lingering use of generation.

7. PLOS authors have the option to publish the peer review history of their article (what does this mean?). If published, this will include your full peer review and any attached files.

Reviewer #1: No

Reviewer #2: No

---

## [Author Response · Author response to Decision Letter 1]

21 Oct 2021

Our responses are summarized in an attached file.

---

## [Decision Letter · Decision Letter 2]

18 Nov 2021

Improved log-Gaussian approximation for over-dispersed Poisson regression: application to spatial analysis of COVID-19

PONE-D-21-12939R2

Dear Dr. Murakami,

We’re pleased to inform you that your manuscript has been judged scientifically suitable for publication and will be formally accepted for publication once it meets all outstanding technical requirements.

Kind regards,

Luca Citi, PhD

Academic Editor

PLOS ONE

Additional Editor Comments (optional):

Reviewers' comments:

Reviewer's Responses to Questions

**Comments to the Author**

1. If the authors have adequately addressed your comments raised in a previous round of review and you feel that this manuscript is now acceptable for publication, you may indicate that here to bypass the “Comments to the Author” section, enter your conflict of interest statement in the “Confidential to Editor” section, and submit your "Accept" recommendation.

Reviewer #1: All comments have been addressed

Reviewer #2: All comments have been addressed

2. Is the manuscript technically sound, and do the data support the conclusions?

Reviewer #1: Yes

Reviewer #2: Yes

3. Has the statistical analysis been performed appropriately and rigorously? 

Reviewer #1: Yes

Reviewer #2: Yes

4. Have the authors made all data underlying the findings in their manuscript fully available?

Reviewer #1: Yes

Reviewer #2: Yes

5. Is the manuscript presented in an intelligible fashion and written in standard English?

Reviewer #1: Yes

Reviewer #2: Yes

6. Review Comments to the Author

Reviewer #1: The authors have adequately addressed all the comments that raised in a previous round of review. o o o o o o o o o o o o o o o o o o o o o o o o o o o o o o o o o o o o o o o o o o o o o o o o o o o o o o o o o o o o o o o o

Reviewer #2: Thank you again for a substantial revision that has improved the manuscript. You have addressed all my concerns.

Details:

Line 186 in revision 1: Thank you for the clarification. I had not seen this method of simulating counts with Var = k*mu before. That’s really nice.

7. PLOS authors have the option to publish the peer review history of their article (what does this mean?). If published, this will include your full peer review and any attached files.

Reviewer #1: **Yes: **Zakariya Y. Algamal

Reviewer #2: No

---

## [Editor Report · Acceptance letter]

23 Dec 2021

PONE-D-21-12939R2 

Improved log-Gaussian approximation for over-dispersed Poisson regression: application to spatial analysis of COVID-19 

Dear Dr. Murakami:

I'm pleased to inform you that your manuscript has been deemed suitable for publication in PLOS ONE. Congratulations! Your manuscript is now with our production department. 

Kind regards, 

on behalf of

Dr. Luca Citi 

Academic Editor

PLOS ONE